# Antiviral Activity of Probenecid and Oseltamivir on Influenza Virus Replication

**DOI:** 10.3390/v15122366

**Published:** 2023-11-30

**Authors:** Jackelyn Murray, David E. Martin, Fred D. Sancilio, Ralph A. Tripp

**Affiliations:** 1Department of Infectious Disease, College of Veterinary Medicine, University of Georgia, Athens, GA 30602, USA; jcrab@uga.edu; 2TrippBio, Inc., Jacksonville, FL 32256, USA; davidmartin@trippbio.com (D.E.M.); fredsancilio@clearwayglobal.com (F.D.S.)

**Keywords:** antiviral, influenza, neuraminidase inhibitor, OAT3 inhibitor, respiratory epithelial cell, BALB/c mice

## Abstract

Influenza can cause respiratory infections, leading to significant morbidity and mortality in humans. While current influenza vaccines offer varying levels of protection, there remains a pressing need for effective antiviral drugs to supplement vaccine efforts. Currently, the FDA-approved antiviral drugs for influenza include oseltamivir, zanamivir, peramivir, and baloxavir marboxil. These antivirals primarily target the virus, making them vulnerable to drug resistance. In this study, we evaluated the efficacy of the neuraminidase inhibitor, oseltamivir, against probenecid, which targets the host cells and is less likely to engender resistance. Our results show that probenecid has superior antiviral efficacy compared to oseltamivir in both in vitro replication assays and in vivo mouse models of influenza infection.

## 1. Introduction

An improved understanding of virus replication has led to the identification of antiviral drugs that target the virus or essential host cell functions necessary for replication. RNA interference (RNAi) silences host genes and can reveal pro- and/or antiviral host genes that affect replication. RNAi studies from our group discovered the key host genes used by viruses to replicate in human respiratory epithelial (A549) cells [1,2,3,4,5]. Using high throughput whole genome screening and RNA interference (RNAi), we pinpointed several crucial host genes whose loss of function led to the inhibition of influenza replication. The RNAi screens used human A549 cells infected with A/WSN/33 and other influenza strains to reveal the key host genes needed for virus replication. A notable gene discovered as important for influenza virus replication was the organic anion transporter-3 gene (OAT3), which is a member of the SLC22 family. As OAT3 is required for influenza virus replication, we tested a drug that inhibited OAT3, i.e., probenecid. OAT3 was necessary for influenza virus replication because the RNAi knockdown of OAT3 prevented virus replication. Thus, we investigated a drug that inhibits OAT3 in host cells and has no effect on the virus, i.e., probenecid. Probenecid is an FDA-approved and safe drug with a >7 decade safety profile for treating gout [6]. The probenecid treatment of infected or uninfected A549 cells inhibited OAT3 mRNA and reduced protein levels, and in vitro or in vivo treatment substantially inhibited viral lung titers in a BALB/c mouse model. These studies show that probenecid reduces influenza A virus (IAV) replication in vitro (IC_50_ = 5 × 10^−5^ − 5 × 10^−4^ μM; *p* < 0.05) and in mice treated daily for over 3 days with 25 mg/kg probenecid as well as protecting from lethal challenge with mouse-adapted A/WSN/33 (60% survival; *p* < 0.05) [7]. These studies also show that the RNAi silencing of closely related transporters, i.e., OAT1, OAT2, OAT4, OAT7, or URAT1, did not affect IAV replication, indicating that OAT3 supported IAV replication. Additionally, the studies show that probenecid prophylaxis or treatment inhibits A/WSN/33, A/New Caledonia/20/99, A/California/07/09, and A/Philippines/2/82/X-79 replication in A549 cells and BALB/c mice. Mice treated prophylactically with 200 mg/kg of probenecid (24 h pre-infection), therapeutically with 200 mg/kg probenecid (24 hpi), or administered 25 mg/kg of probenecid daily for 3 days following infection had reduced morbidity and mortality and low-to-no lung virus titers.

OAT3 is a member of the solute carrier (SC) family consisting of several OATs and OAT-like members [8]. SC transporters mediate the translocation of substrates across membranes [9]. Probenecid has recently been shown to have in vitro and in vivo antiviral activity for SARS-CoV-2 and the respiratory syncytial virus (RSV) [7,10,11,12]. In these studies, it was shown that probenecid prophylaxis or treatment inhibited SARS-CoV-2 replication as well as several variants of concern, e.g., Beta, Gamma, Delta, and B.1.1 in Vero E6 cells and normal human bronchial epithelial (NHBE) cells at submicromolar concentrations, i.e., 0.00001–100 μM. Hamsters were treated with probenecid 24 h before infection or 48 h post-infection with 2 or 200 mg/kg probenecid and had dramatically reduced lung virus titers, i.e., a 4–5_log_ reduction in the virus compared to the controls that were approximately 10^9^ logs of the virus [8]. In addition, probenecid treatment reduced RSV replication in three strains, i.e., strain A2, A/Memphis-37, and B1, in human respiratory epithelial cell lines and mice. Maximum reductions in the lung virus load occurred in mice pretreated with 200 mg/kg of probenecid 24 h before infection. Mice therapeutically treated once with 2 or 200 mg/kg of probenecid 24 h after RSV infection also had greatly reduced RSV A2 lung titers on days 3, 5, and 7 pi. Maximum reductions in the lung virus load occurred in mice treated with 200 mg/kg probenecid. Notably, results from a Phase 2 randomized, placebo-controlled, single-blind, dose-range finding study in non-hospitalized patients with symptomatic, mild-to-moderate COVID-19 showed that 1000 mg of probenecid treatment significantly reduced the time to viral clearance in patients compared to the placebo (7 days vs. 11 days, respectively; *p* < 0.0001), and for patients treated with the 500 mg probenecid group versus placebo (9 days vs. 11 days, respectively; *p* < 0.0001). In addition, the median time to viral clearance was significantly shorter for the probenecid 1000 mg group than for the probenecid 500 mg group (7 days vs. 9 days, respectively; *p* < 0.0001) [12].

Given the pan-antiviral effects of probenecid, this mechanism of action was addressed because it likely involved a host cell pathway used by more than one virus for replication. Several host factors have been associated with RSV replication, including importin-1, Crm1, cofilin, caveolin, ZNF502, and the serine/threonine protein kinase CK2 [13,14,15,16,17]. In addition, c-Jun N-terminal kinase (JNK) activity is known to be required for RSV replication [18]. JNK is an important component in MAPK signaling and becomes activated by a virus infection [19]. JNK activates MAPK, downregulating the expression of hepatocyte nuclear factor-4 alpha (HNF-4), and HNF-4 regulates the expression of certain OATs [8,20,21]. Emerging data suggest that the mechanism of action of probenecid involves inhibiting JNK phosphorylation and the downstream HNF-4 regulation of OAT3, likely inhibiting virus assembly and replication. Recent studies suggest that probenecid might also affect other MAP kinases and possibly extracellular signal-regulated kinases (ERKs). The inhibition of JNK phosphorylation also inhibits the JNK activator protein-1 (AP-1) that regulates the transcription of pro-inflammatory cytokines (IL-1, IL-6, TNF) and inflammatory enzymes (e.g., COX-2) [22]. Probenecid has also been shown to reduce ACE2 expression [23] and modify pannexin 1 (PANX1) gene expression [24], which regulates inflammation and host responses to viruses as well as the NLRP3 inflammasome response associated with severe influenza virus infection [25]. Probenecid can also reduce inflammation mediated through the P2X7 receptor and increase bacterial clearance in a murine model of *P. aeruginosa* pneumonia without a direct antibacterial effect [26].

Influenza antivirals are divided into the following two categories of direct-acting drugs: neuraminidase (NA) inhibitors (NAIs) that target the NA enzyme to prevent virus budding and the cap-dependent endonuclease inhibitor, baloxavir marboxil. While baloxavir is primarily used in Japan [27], NAIs have global availability. Presently, there are four NAIs as follows: oseltamivir (oral), zanamivir (inhaled) and peramivir (intravenous)—which are approved in China—and laninamivir octanoate (inhaled), which is exclusively approved in Japan [28]. Oseltamivir is a prodrug of oseltamivir carboxylate and a selective inhibitor of NA. Oseltamivir (Tamiflu™, F. Hoffmann-La Roche Ltd, Basel, Switzerland) was approved by the FDA in 1999 as an oral treatment for uncomplicated influenza [27]. It inhibits the activity of NA and prevents the replication of influenza A and B [28]. It has been shown that early treatment with oseltamivir in adult hospitalized patients can shorten fever and hospitalization [29]. Importantly, oseltamivir is effective against both A and B virus strains. It can be used for a 1- to 2-day treatment window from the onset of symptoms, and it has a rapid onset of action, yielding clinically significant reductions in the duration and severity of symptoms [30]. Current strains of influenza have low drug resistance, but this may change. Studies have shown that a mutation in NA that converts histidine 275 to tyrosine (H275Y) confers resistance to oseltamivir in A(H1N1) pdm09 flu viruses [31]. Thus, we compared the antiviral efficacy of probenecid to oseltamivir in a preclinical model. The IC_50_ of probenecid against the influenza A infection in vitro for H1N1 A/WSN/33 and A/New Caledonia/20/99 ranged from 5 × 10^−5^ to 5 × 10^−4^ μM, while the IC_50_ of oseltamivir for A(H1N1) pdm09 isolates was 0.13–0.15 μM [32].

## 2. Materials and Methods

### 2.1. Cell Lines and Viruses

Human type II respiratory epithelial A549 cells (ATCC; CCL-185, Manassas, VA, USA) and Madin–Darby canine kidney (MDCK) cells (ATCC; CCL-34) were cultured in Dulbecco’s modified Eagle’s medium (DMEM) supplemented with 5% heat-inactivated fetal bovine serum (FBS) (HyClone, Logan, UT, USA) and grown in a 37 °C incubator with 5% CO_2_. Normal human bronchial epithelial (NHBE) cells (LifeLine Cell Technology, Frederick, MD, USA) were cultured and maintained. NHBE cells are primary human cells and were maintained at an air–liquid interface at 37 °C with 5% CO_2_.

Influenza virus strains A/Mississippi/3/2001 (H1N1, A/New Caledonia/20/99-like), A/Mississippi/3/2001 H275Y (from ISIRV Antiviral Group), and influenza B virus (Yamagata Lineage, ATCC VR-1804), were propagated in 9-day-old embryonic chicken eggs, and titers in MDCK cells were determined. Titers were determined via a plaque assay on MDCK cells in the presence of 5% FBS at 37 °C.

### 2.2. Antiviral Drugs and In Vitro Assays

Probenecid (CAS Number: 57-66-9) (Invitrogen, Carlsbad, CA, USA) was diluted in DMSO (Sigma, St. Louis, MO, USA) and resuspended in PBS (Gibco, ThermoFisher, Waltham, MA, USA) and oseltamivir carboxylate (Sigma, St. Louis, MO, USA); the active metabolite of oseltamivir phosphate (Tamiflu) was examined in vitro for its inhibitory effect on A/Mississippi/3/2001, A/Mississippi/3/2001 H275Y, or B/Florida/4/2006 replication in A549 cells or NHBE cells [33]. A/Mississippi/3/2001 is a wild-type influenza virus with a histidine at position 275 (275H) of the NA, while A/Mississippi/3/2001 H275Y is a variant with a tyrosine at position 275 (a H275Y substitution) of NA. This variant is known to be resistant to oseltamivir (https://www.isirv.org/site/index.php/mutant-characteristics, accessed on 1 October 2023).

A549 cells or NHBE cells were plated overnight at 10^4^ cells/well in 96-well flat-bottom plates (Costar). Cells were pretreated for 24 h before infection (prophylactically) or therapeutically at 1 hpi with probenecid or oseltamivir at different concentrations, i.e., 100,000, 10,000, 1000, 100, 10, 1, 0.1, 0.01, 0.001 or 0 μM. For prophylactically treated cells, the media and probenecid were removed, and the cells were infected with the A/Mississippi/3/2001, A/Mississippi/3/2001 H275Y, or B/Florida/4/2006 at MOI = 0.1. At 72 hpi, the virally infected cells were fixed with cold methanol–acetone and subsequently anti-NP immunostaining, as described [34].

### 2.3. BALB/c Mouse Assays

BALB/c female mice (6–8 weeks old) were obtained from Charles River (Raleigh, NC, USA) and were rested a week before use. The animal study protocol was approved by the Institutional Review Board of the University of Georgia, A2021 03-006-Y2-A0, Immunity to Respiratory Viruses and Virus Proteins in Mus musculus, approved on 6 May 2021. In all experiments evaluating the antiviral efficacy of probenecid relative to oseltamivir carboxylate, BALB/c mice were i.n. infected (MOI = 1) with A/Mississippi/3/2001, A/Mississippi/3/2001 H275Y, or B/Florida/4/2006. The groups (n = 5) of mice were administered one of six different treatment regimens as follows: (1) a probenecid pro-drug (probenecid triethylene glycol ester, PTGE) at 10 mg/kg, (2) PTGE at 100 mg/kg, (3) probenecid at 100 mg/kg, (4) a vehicle only, (5) oseltamivir phosphate at 10 mg/kg, or (6) mice were uninfected and untreated (control). All test articles were administered via oral gavage (OG). PTGE OG was directly compared to oseltamivir at 10 mg/kg or 100 mg/kg via oral gavage twice daily every 12 h until day 5 pi. The dose of oseltamivir was selected to achieve the same area-under-the-curve pharmacokinetic profile of orally administered prodrug oseltamivir phosphate in humans [28]. The NA-H275Y substitution is highly clinically significant because seasonal A(H1N1) variants with the NA-H275Y substitution spread globally in the 2007–2008 influenza season [35]. Lungs were processed and titered on MDCK cells at day 5 pi.

### 2.4. Probenecid Pro-Drug

The probenecid pro-drug, probenecid triethylene glycol ester (PTGE), was prepared by Quality Chemical Laboratory (Wilmington, DE, USA).

### 2.5. Statistical Analysis

The results were analyzed using one-way ANOVA followed by Turkey’s post hoc test for multiple comparisons of the control and treatment groups or Student’s t-test using GraphPad Prism (version 6). IC_50_ and IC_90_ values were calculated from GraphPad Prism (GraphPad Software, La Jolla, CA, USA).

## 3. Results

RNAi screens of A549 cells infected with H1N1 A/WSN/33 showed the requirement of OAT3 for influenza replication [2,4,6]. Additionally, the transfection of A549 cells with siRNA targeting OAT3 completely blocked influenza A/WSN/33 virus replication, and probenecid treatment inhibited the A/WSN/33 replication in vitro (IC_50_) of 5 × 10^−^^5^ − 5 × 10^−^^4^ μM [6] and reduced OAT3 mRNA and protein levels in vitro and in BALB/c mice.

To compare the antiviral effects of probenecid to oseltamivir, studies in A549 cells and NHBE cells were examined. Specifically, A/Mississippi/3/2001, A/Mississippi/3/2001 H275Y, or B/Florida/4/2006 drug sensitivity was determined. A549 cells were either (A) prophylactically treated 24 h before A/Mississippi/3/2001 infection or (B) treated 1 h after infection with A/Mississippi/3/2001. Ten-fold dilutions of probenecid or oseltamivir (100,000, 10,000, 1000, 100, 1, 0.1, 0.01, 0.001, or 0 μM) were examined. The IC_50_/IC_90_ for prophylactic treatment with probenecid was 0.0002 μM/0.1947 μM and 0.0014 μM/0.8968 μM for oseltamivir treatment. The IC_50_/IC_90_ for drug treatment 1 h after infection with probenecid was 0.001 μM/0.255 μM, and 0.045 μM/3.6 μM for the IC_50_/IC_90_ for the oseltamivir treatment of A/Mississippi/3/2001-infected cells (Figure 1). These findings show that probenecid reduced the viral load better than oseltamivir.

To determine the effect of drug resistance on the antiviral activity profile, A549 cells were infected with oseltamivir-resistant A/Mississippi/3/2001 H275Y [36]. The A549 cells were prophylactically treated with probenecid or oseltamivir. Ten-fold dilutions of probenecid or oseltamivir (100,000, 10,000, 1000, 100, 1, 0.1, 0.01, 0.001, or 0 μM) were examined. The IC_50_/IC_90_ for probenecid was 0.0002 μM/0.0018 μM, and for oseltamivir, it was 0.013 μM/1.7 μM. The IC50/IC_90_ for probenecid treatment 1 h after infection was 0.0009 μM/0.251 μM, and for oseltamivir, it was 0.049 μM/9.2 μM (Figure 2). Probenecid reduced the viral load better than oseltamivir for both treatments.

NHBE cells are a primary cell line derived from human biopsies of the trachea, while A549 cells are lung carcinoma epithelial cells that constitute a cell line. Both cell types are readily infected by influenza viruses and represent good models of influenza infection [37,38]. To determine the antiviral effects of probenecid or oseltamivir on NHBE cells, 10-fold dilutions of probenecid or oseltamivir (100,000, 10,000, 1000, 100, 1, 0.1, 0.01, 0.001 or 0 μM) were examined. The cells were infected with A/Mississippi/3/2001 (Figure 3). The IC_50_/IC_90_ for the prophylactic treatment of NHBE cells with probenecid was 0.005 μM/0.121 μM, and for oseltamivir, it was 0.0348 μM/2.82 μM. The IC_50_/IC_90_ for probenecid treatment 1 h after infection was 0.00049 μM/0.313 μM and 0.172 μM/3.2 μM for oseltamivir-treated NHBE cells infected with A/Mississippi/3/2001 (Figure 3). Probenecid reduced the viral load better than oseltamivir for both drug treatments.

To determine the antiviral effectiveness in a drug-resistant virus, NHBE cells were infected with A/Mississippi/3/2001 H275Y and treated with either probenecid or oseltamivir (Figure 4). Ten-fold dilutions of probenecid or oseltamivir (100,000, 10,000, 1000, 100, 1, 0.1, 0.01, 0.001, or 0 μM) were examined. The IC_50_/IC_90_ for the prophylactic treatment of NHBE cells with probenecid was 0.595 μM/16.6 μM, and the IC_50_/IC_90_ for oseltamivir treatment was 1.2 μM/27.6 μM. The IC_50_/IC_90_ 1 h after infection was 0.0059 μM/3.17 μM for probenecid and 0.1511 μM/12.2 μM for oseltamivir-treated NHBE cells infected with A/Mississippi/3/2001(Figure 4). Probenecid reduced the viral load better than oseltamivir for both treatments.

Having shown the antiviral effects of influenza A-infected human respiratory cells, the antiviral effects of probenecid and oseltamivir were determined for Yamagata-like B/Florida/4/2006 A549 infected cells (Figure 5). A549 cells were either (A) prophylactically treated 24 h before infection with B/Florida/4/2006 or (B) treated 1 h after infection with B/Florida/4/2006. Ten-fold dilutions of probenecid or oseltamivir (100,000, 10,000, 1000, 100, 1, 0.1, 0.01, 0.001, or 0 μM) were examined. The IC_50_/IC_90_ for the prophylactic treatment of A549 cells with probenecid was 0.0014 μM/82 μM, and the IC_50_/IC_90_ for the oseltamivir treatment was 0.010.6 μM/625 μM. The IC_50_/IC_90_ for treatment 1 h after infection with probenecid was 0.0486 μM/73 μM, and for oseltamivir was 0.023 μM/36 μM (Figure 5). Probenecid reduced the viral load better than oseltamivir for both treatments.

To evaluate drug efficacy in NHBE cells, the cells were either (A) prophylactically treated 24 h before infection with B/Florida/4/2006 or (B) treated 1 h after infection with B/Florida/4/2006. Ten-fold dilutions of probenecid or oseltamivir (100,000, 10,000, 1000, 100, 1.0, 0.1, 0.01, 0.001, or 0 μM) were examined. The IC_50_/IC_90_ for prophylactic treatment of A549 cells with probenecid was 0.014 μM/21 μM, and the IC_50_/IC_90_ for the oseltamivir treatment was 0.428 μM/191.5 μM. The IC_50_/IC_90_ for treatment 1 h after infection with probenecid was 0.637 μM/68 μM, and for oseltamivir, it was 3.4 μM/281.3 μM (Figure 6). Probenecid reduced the viral load better than oseltamivir for both treatments.

The IC_50_ values for A/Mississippi/3/2001, A/Mississippi/3/2001 H275Y, and B/Florida/4/2006 following either prophylaxis or treatment with probenecid or the oseltamivir treatment of A5549 cells or NHBE cells is summarized in Table 1.

The fold difference in IC_50_ between probenecid and oseltamivir is remarkable. The effectiveness of probenecid was superior to oseltamivir in all but A549 cells infected with B/Florida/4/2006. Interestingly, the finding that a NA mutation (A/Mississippi/3/2001 H275Y) can impact probenecid sensitivity (IC_50_ = 0.0595 μM) compared to wildtype (A/Mississippi/3/2001; IC_50_ = 0.0005 μM) in NHBE-infected cells is notable; however, the mechanism for this is unknown. Clearly, the overall robustness of probenecid’s effectiveness is overall greater than oseltamivir.

We examined the in vivo inhibitory effect of probenecid and oseltamivir carboxylate on A/Mississippi/3/2001, A/Mississippi/3/2001 H275Y, and B/Florida/4/2006 replication in the lungs of BALB/c mice (Figure 7). A/Mississippi/3/2001 has a histidine at position 275 (275H) of the neuraminidase glycoprotein, while A/Mississippi/3/2001 H275Y is a variant virus that is oseltamivir-resistant and has tyrosine at position 275 (275Y) of neuraminidase. Probenecid is water-insoluble but soluble in 100 mM of DMSO and 100 mM of ethanol. Oseltamivir phosphate (Tamiflu^®^) is an oral antiviral neuraminidase inhibitor that works by stopping the spread of the influenza virus in the body. It is approved for treating acute, uncomplicated influenza in patients 2 weeks of age and older whose symptoms have not lasted more than two days (https://www.ncbi.nlm.nih.gov/books/NBK539909/, accessed on 24 September 2022). Given the insolubility of probenecid, we evaluated a probenecid prodrug, PTGE, which is a probenecid triethylene glycol ester that distributes probenecid throughout the body, mostly as a plasma protein bound predominantly to albumin. In these studies, PTGE OG (oral gavage) was directly compared to oseltamivir at 10 mg/kg or 100 mg/kg via oral gavage treatment twice daily every 12 h until day 5 pi.

The FDA-approved dose for oseltamivir is 75 mg taken BID for 5 days. Mouse studies used doses of 10 mg/kg, while the PTGE (probenecid) doses were either 10 or 100 mg/kg, taken twice daily every 12 h until day 5 pi. The 10 mg/kg/day oseltamivir dose in mice was chosen because the oral bioavailability is similar to the recommended human oral dose of 75 mg BID two times a day [39,40,41]. Additionally, the oseltamivir dosage was adjusted for the interspecies difference in esterase activity and metabolic rates [40]. Oseltamivir had little impact on reducing lung virus titers (PFU/mL) compared to vehicle-treated mice (Figure 7). These findings are similar to earlier studies of lung viral titers in post-infection dosing studies quantitated using an MDCK plaque assay [40]. Comparing oseltamivir treatment to the PTGE dosing groups, it was clear that PTGE OG dosing groups both reduced the viral load better than oseltamivir and PTGE OG 100 mg/kg and probenecid OG 100 mg/kg reduced the viral loads completely in A/Mississippi/3/200-infected mice. However, the probenecid/PTGE compounds were less effective against A/Mississippi/3/2001 H275Y than wild-type viruses. The weights of the BALB/c mice infected with A/Mississippi/3/2001, A/Mississippi/3/2001 H275Y, or B/Florida/4/2006 were determined. Mice were weighed before infection and every day following infection. The mice in the vehicle control group lost the highest percentage of body weight in all of the studies. There were no animals that lost 20% of their body weight before day 5 pi. Oseltamivir and PTGE dosing groups had a modest impact on weight loss due to A/Mississippi/3/2001, A/Mississippi/3/2001 H275Y, or B/Florida/4/2006 infection.

## 4. Discussion

Oseltamivir’s antiviral activity is directed at the viral NA enzyme, which prevents budding from the host cell, viral replication, and infectivity [42]. Oseltamivir is being considered for treating acute, uncomplicated influenza A or B infection in patients who have been symptomatic for no more than 48 h [43]. There is uncertainty regarding the use of oseltamivir to treat adults and adolescents with confirmed influenza illnesses. For example, findings from a systematic review and meta-analysis of 15 clinical trials showed that oseltamivir treatment was not associated with a reduced risk of hospitalization compared with a placebo or the standard of care [44]. Oseltamivir reduces the duration of influenza symptoms by only 12–24 h [45]. Some people take oseltamivir prophylactically, but the CDC does not recommend this practice because it may lead to the development of resistance. However, high-risk individuals should take it if they have not been (or have only recently been) vaccinated or exposed to influenza in the previous 48 h. Thus, oseltamivir treatment effectively reduces the duration of symptoms, but evidence for it causing a reduction in transmission remains inconclusive.

Probenecid is an FDA-approved generic medication that has been used for nearly seven decades to treat gout and has an outstanding safety record. Probenecid blocks viral replication by targeting host factors, e.g., OAT3 [6] and JNK and MAPK [46], rather than the virus. Since its effect is on the host cell, it exhibits broad-spectrum antiviral activity and maintains efficacy in the face of mutations and variants [7]. It is likely that the broad antiviral effectiveness of probenecid for SARS-CoV-2, RSV, and influenza is linked to the host genes affected by the treatment [7]. We observed that the antiviral activity was linked to OAT3 [6], and Western blot evidence indicates that probenecid prevents JNK phosphorylation, affecting the c-jun protein and HNF-4 expression [47,48] as a transcription factor that regulates OAT3 expression [49,50].

We showed that probenecid is an effective antiviral drug against several contemporary influenza A and B strains [6] and was effective against an Asian lineage of the avian influenza A (H7N9) virus. Here, we show that probenecid is better at reducing the influenza A virus in A549 cells and NHBE cell studies and in vivo in A/Mississippi/3/2001-infected mice compared to oseltamivir. In addition, the PTGE pro-drug demonstrated the proof-of-concept in the mouse model of influenza infection. These data confirm the potent antiviral effect of probenecid and its pro-drug PTGE. It is interesting that probenecid/PTGE compounds were less effective against A/Mississippi/3/2001 H275Y, suggesting that NA mutations might impact probenecid sensitivity; however, the mechanism for this possibility is unknown.

## Figures and Tables

**Figure 1 viruses-15-02366-f001:**
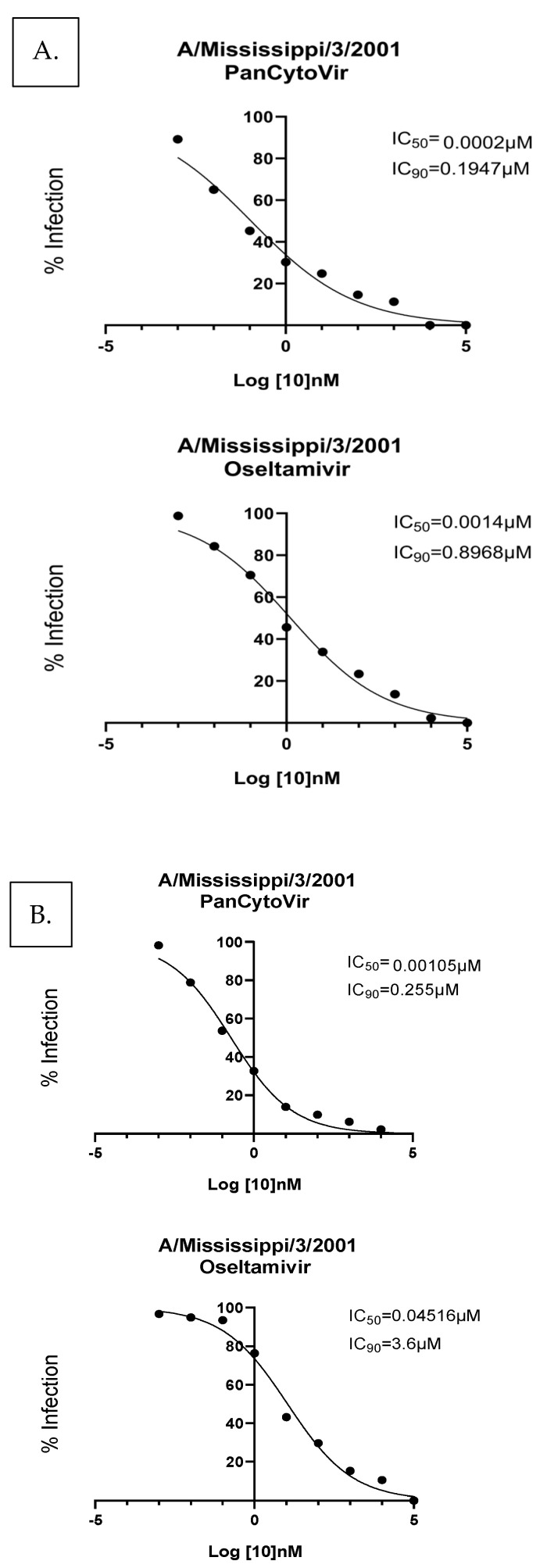
A549 cells were (**A**) prophylactically treated 24 h before A/Mississippi/3/2001 infection, or (**B**) treated 1 h after A/Mississippi/3/2001 infection. A549 cells were infected at a MOI of 0.1 and incubated for 1 h. After incubation, the infection was removed, and the cells were washed. Media containing PanCytoVir or Oseltamivir was added to the cells. These data represent three independent experiments, each with three experimental replicates.

**Figure 2 viruses-15-02366-f002:**
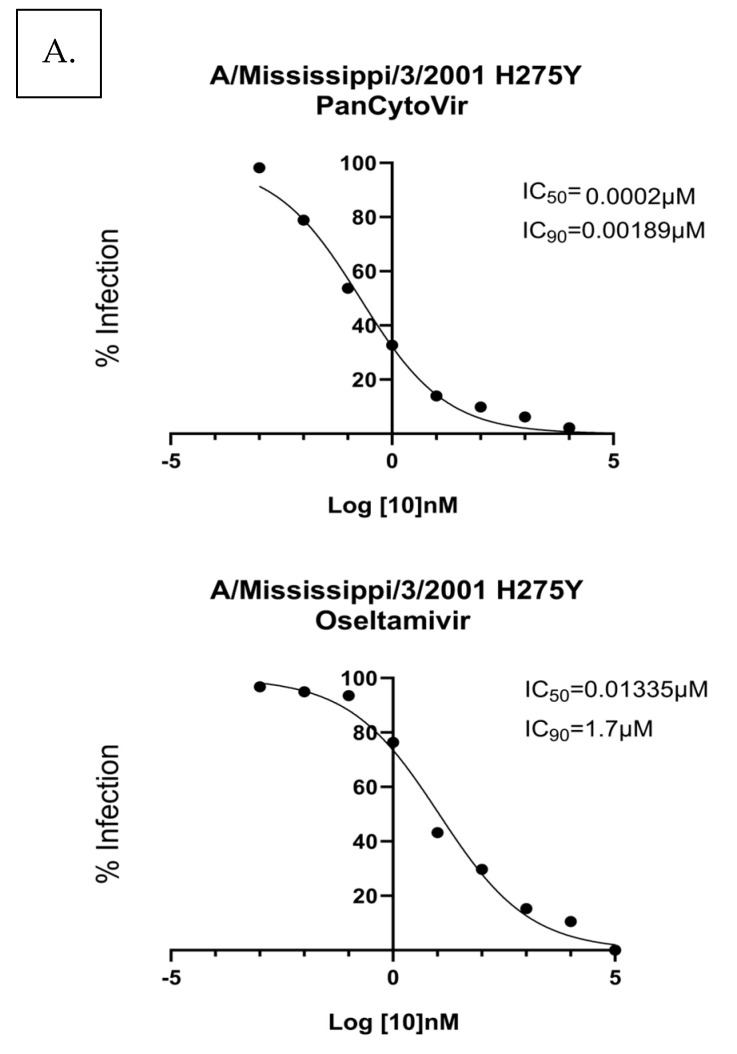
A549 cells were (**A**) prophylactically treated 24 h before infection with A/Mississippi/3/2001 H275Y or (**B**) treated 1 h after infection with A/Mississippi/3/2001 H275Y. A549 cells were infected at a MOI of 0.1 and incubated for 1 h. After incubation, the infection was removed, and the cells were washed. Media containing PanCytoVir or Oseltamivir was added to the cells. These data represent three independent experiments, each with three experimental replicates.

**Figure 3 viruses-15-02366-f003:**
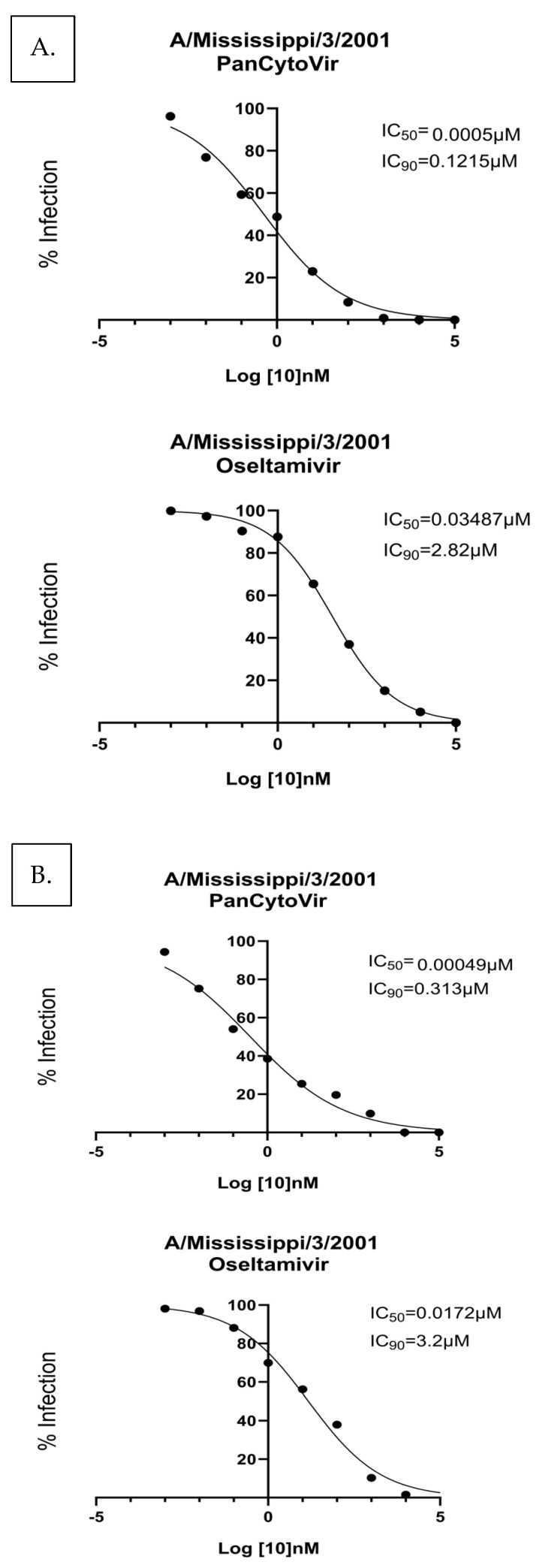
NHBE cells were (**A**) prophylactically treated 24 h prior to infection with A/Mississippi/3/2001, or (**B**) treated 1 h after infection with A/Mississippi/3/2001. NHBE cells were infected at a MOI of 0.1 and incubated for 1 h. After incubation, the infection was removed, and the cells were washed. Media containing PanCytoVir or Oseltamivir was added to the cells. These data represent 3 independent experiments, each with three experimental replicates.

**Figure 4 viruses-15-02366-f004:**
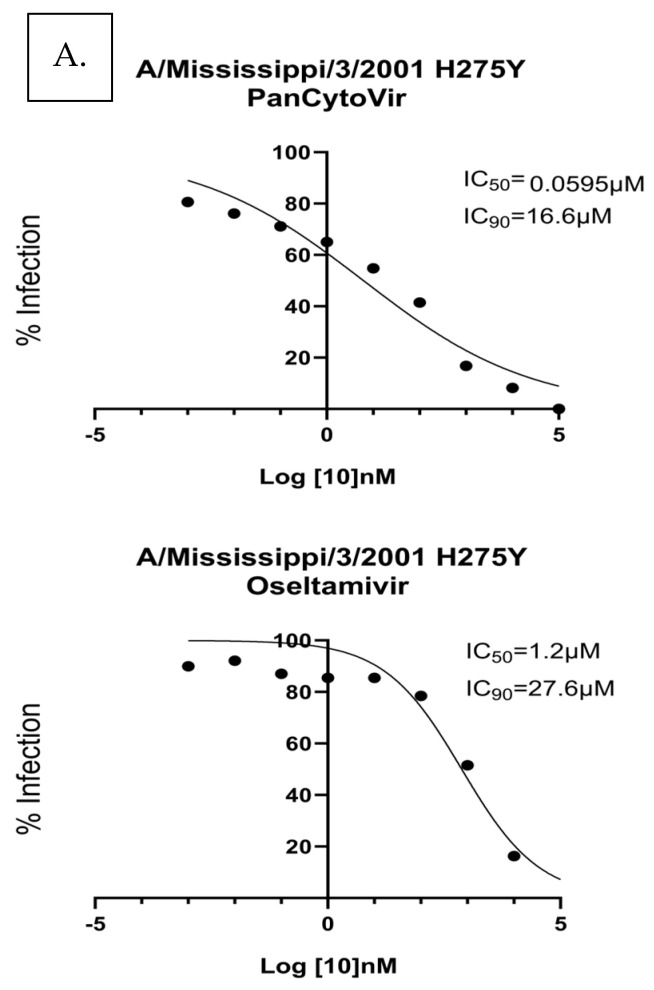
NHBE cells were (**A**) prophylactically treated 24 h prior to infection with A/Mississippi/3/2001 H275Y or (**B**) treated 1 h after infection with A/Mississippi/3/2001 H275Y. NHBE cells were infected at a MOI of 0.1 and incubated for 1 **h**. After incubation, the infection was removed, and the cells were washed. Media containing PanCytoVir or Oseltamivir was added to the cells. These data represent 3 independent experiments, each with three experimental replicates.

**Figure 5 viruses-15-02366-f005:**
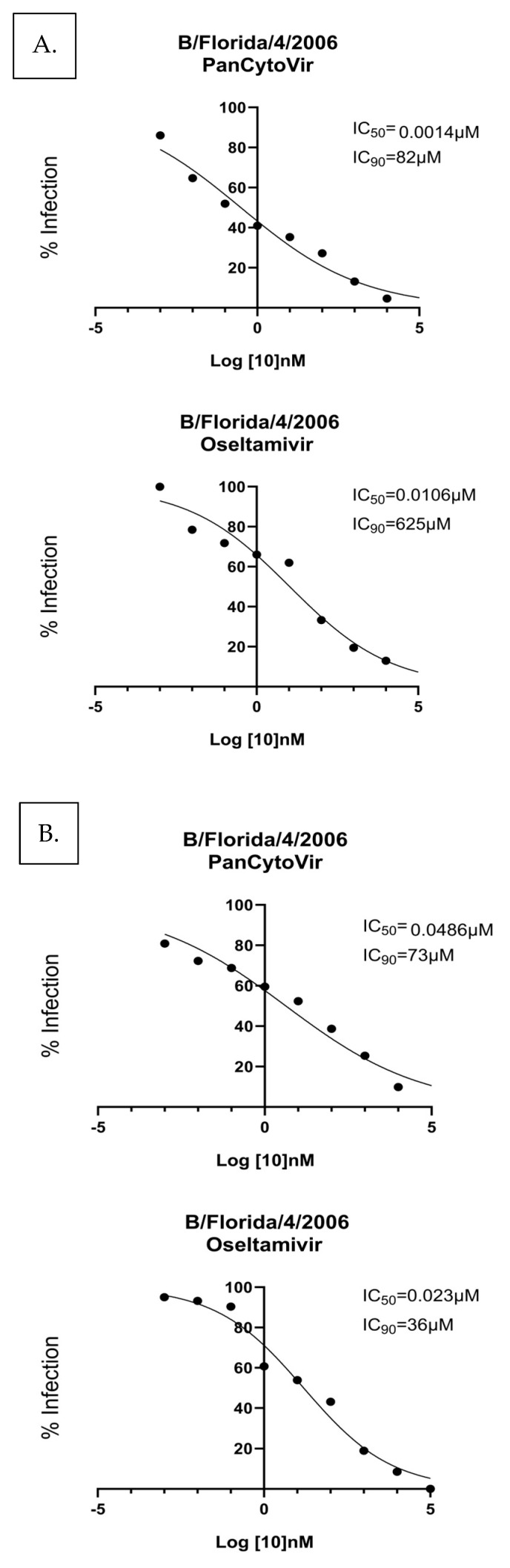
A549 cells were (**A**) prophylactically treated 24 h before infection with B/Florida/4/2006, or (**B**) treated 1 h after infection with B/Florida/4/2006. A549 cells were infected at a MOI of 0.1 and incubated for 1 h. After incubation, the infection was removed, and the cells were washed. Media containing PanCytoVir or Oseltamivir was added to the cells. These data represent three independent experiments, each with three experimental replicates.

**Figure 6 viruses-15-02366-f006:**
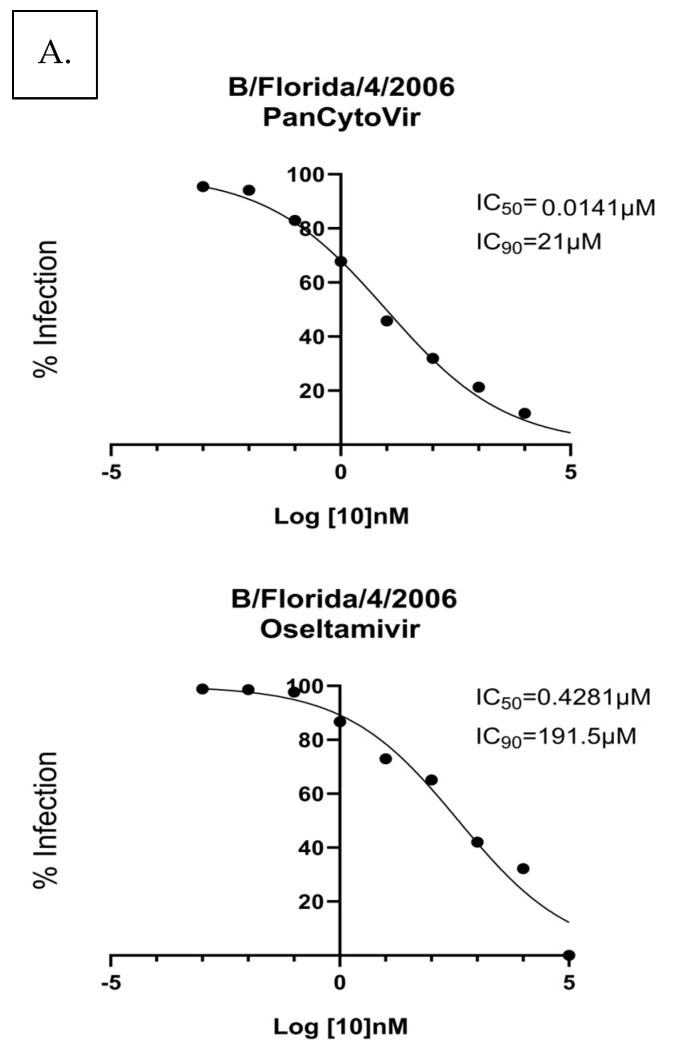
NHBE cells were (**A**) prophylactically treated 24 h prior to infection with B/Florida/4/2006 or (**B**) treated 1 h after infection with B/Florida/4/2006/NHBE cells were infected at an MOI of 0.1 and incubated for 1 h. After incubation, the infection was removed, and the cells were washed. Media containing PanCytoVir or Oseltamivir was added to the cells. These data represent three independent experiments, each with three experimental replicates.

**Figure 7 viruses-15-02366-f007:**
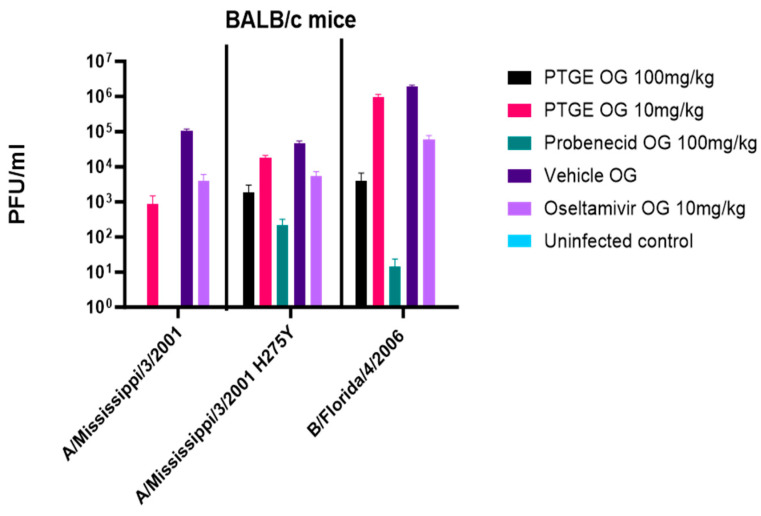
Female 6–8-week-old BALB/c mice were i.n. infected with A/Mississippi/3/2001, A/Mississippi/3/2001 H275Y, or B/Florida/4/2006. Lungs were processed and titered on MDCK cells at day 5 pi. Probenecid reduced the lung’s viral load the greatest compared to oseltamivir. The PTGE OG dosing groups both reduced the viral load better than oseltamivir. PTGE OG 100 mg/kg and probenecid OG 100 mg/kg reduced the viral loads completely in A/Mississippi/3/2001-infected mice.

**Table 1 viruses-15-02366-t001:** Antiviral susceptibility of wildtype and influenza variants

Viral Strain	Cell Line	Probenecid (IC_50_; μM)	Oseltamivir (IC_50_; μM)	Fold Difference
Prophylaxis				
A/Mississippi/3/2001	A549	0.0002	0.0014	7
	NHBE	0.0005	0.0345	69
A/Mississippi/3/2001 H275Y	A549	0.0002	0.0133	66
	NHBE	0.0595	1.2	20
B/Florida/4/2006	A549	0.0014	0.0106	7
	NHBE	0.0141	0.428	30
Treatment				
A/Mississippi/3/2001	A549	0.0009	0.0451	50
	NHBE	0.0049	0.0172	35
A/Mississippi/3/2001 H275Y	A549	0.0009	0.0494	55
	NHBE	0.0059	0.151	25
B/Florida/4/2006	A549	0.0486	0.023	0.5
	NHBE	0.637	34.0	53

## Data Availability

The data supporting the reported results can be found in the Tripp laboratory at the University of Georgia.

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
