# Peer review of "Antiviral Activity of Probenecid and Oseltamivir on Influenza Virus Replication"

_viruses, 2023, doi:10.3390/v15122366_

Round 1

Reviewer 1 Report

Comments and Suggestions for Authors

The article shows the antiviral activity of probenecid against some influenza A and B strains making a comparative analysis with the anti-flu drug oseltamivir. Probenecid exhibited a greater inhibition potency against the investigated virus strains than oselltamivir, also the H275Y drug resistant mutant virus. The antiviral screening was based on in vitro and in vivo tests.

The paper is interesting, the English style is good. References are adequate.

My suggestions are as follows:

In the introduction, the Authors should re-organise the biological properties adding an appropriate Figure, to make the article reading easier.

Since probenecid works as an host targeting antiviral, this relevant concept should be more stressed and dealth with in the introduction and in the discussion. 

Comments on the Quality of English Language

The Englysh style is good.

Author Response

Reviewer 1:

The article shows the antiviral activity of probenecid against some influenza A and B strains making a comparative analysis with the anti-flu drug oseltamivir. Probenecid exhibited a greater inhibition potency against the investigated virus strains than oselltamivir, also the H275Y drug resistant mutant virus. The antiviral screening was based on in vitro and in vivo tests.

The paper is interesting, the English style is good. References are adequate. My suggestions are as follows:

In the introduction, the Authors should re-organise the biological properties adding an appropriate Figure, to make the article reading easier.

Since probenecid works as an host targeting antiviral, this relevant concept should be more stressed and dealth with in the introduction and in the discussion.

  • We appreciate the reviewer’s comments and concur, and now emphasize the activity of probenecid to be on the target cell in the Introduction, and it is noted in the Discussion on lines 358-359. We have reorganized the Introduction to facilitate readability. We think such reorganization obviates the need for a Figure.

Reviewer 2 Report

Comments and Suggestions for Authors

The manuscript "Antiviral Activity of Probenecid or Oseltamivir on Influenza Replication” by Murray et al., is an apparently logical effort from the Tripp group. In it, the authors compared the two antivirals mentioned in the title, for their efficacy as antivirals against influenza virus (The conjunction “or” in the title dose not make much sense”).

The pharmaceutical use of Probenecid, which the major focus in this paper, has been primarily in the treatment of gout and related arthritis, as it reduces the uric acid levels in the blood by promoting the excretion of uric acid in the urine. Its test as an antiviral is relatively recent, and Dr. Tripp has cited ~11 of their own papers about its repurposing in various RNA viruses, which include the first 7 citations, ref 7 (line 430; incompletely cited) being also published in this journal (Viruses). 

The methodology is routine virology, quantifying virus growth by one of several age-old assays that covered multiple treatment regimens (dosage) in mice, and subsequent plaque assay of the lung hydrolysates on MDCK, NHBE and A549 cells. All assays showed a higher antiviral effect of Probenecid over Oseltamivir. The results appear straightforward, not much to review here.

 A notable feature of this paper is its substantial conflict of interest, whereby all the key investigators are employee and /or shareholders of Dr. Tripp’s company, TrippBio (Line 410-411). This is not surprising, since Institutional Research Foundations are generally focused toward Tech Transfer; however, one must hope that the superiority of Probenecid over the currently used Oseltamivir, as reported here, would be replicated by an independent group.

All drugs, targeting a host function that is also co-opted by a virus, has some side-effects, and so does Probenecid, which inhibits OAL3, an important member of the host SLC22 family. Oseltamivir, in contrast, is a Direct-Acting Antiviral (DAA), by virtue of its ability to specifically inhibit the viral neuraminidase (NA) inhibitor, with little effect on host functions, although the trade-off is the selection of Oseltamivir-resistant virus mutants due to mutations in neuraminidase. Thus, Probenecid may only be used in such cases, or in a combination therapy.  

However, the authors have remained vague about the appearance of Probenecid-resistant viral mutants. In the last sentence (line 393-396), the authors speculate that some NA mutations may also exhibit Probenecid-resistance, but it is not clear why this important issue, which could further compare Probenecid with Oseltamivir, was not pursued further.

 Minor: The references are probably not MDPI-formatted. Line 408 needs to corrected.

Author Response

Reviewer 2:

The manuscript "Antiviral Activity of Probenecid or Oseltamivir on Influenza Replication” by Murray et al., is an apparently logical effort from the Tripp group. In it, the authors compared the two antivirals mentioned in the title, for their efficacy as antivirals against influenza virus (The conjunction “or” in the title dose not make much sense”).

  • We have changed the conjugation in the title to “and”.

The pharmaceutical use of Probenecid, which the major focus in this paper, has been primarily in the treatment of gout and related arthritis, as it reduces the uric acid levels in the blood by promoting the excretion of uric acid in the urine. Its test as an antiviral is relatively recent, and Dr. Tripp has cited ~11 of their own papers about its repurposing in various RNA viruses, which include the first 7 citations, ref 7 (line 430; incompletely cited) being also published in this journal (Viruses).

  • We are sorry for the incomplete reference. It is now corrected.

The methodology is routine virology, quantifying virus growth by one of several age-old assays that covered multiple treatment regimens (dosage) in mice, and subsequent plaque assay of the lung hydrolysates on MDCK, NHBE and A549 cells. All assays showed a higher antiviral effect of Probenecid over Oseltamivir. The results appear straightforward, not much to review here.

 A notable feature of this paper is its substantial conflict of interest, whereby all the key investigators are employee and /or shareholders of Dr. Tripp’s company, TrippBio (Line 410-411). This is not surprising, since Institutional Research Foundations are generally focused toward Tech Transfer; however, one must hope that the superiority of Probenecid over the currently used Oseltamivir, as reported here, would be replicated by an independent group.

All drugs, targeting a host function that is also co-opted by a virus, has some side-effects, and so does Probenecid, which inhibits OAL3, an important member of the host SLC22 family. Oseltamivir, in contrast, is a Direct-Acting Antiviral (DAA), by virtue of its ability to specifically inhibit the viral neuraminidase (NA) inhibitor, with little effect on host functions, although the trade-off is the selection of Oseltamivir-resistant virus mutants due to mutations in neuraminidase. Thus, Probenecid may only be used in such cases, or in a combination therapy. 

However, the authors have remained vague about the appearance of Probenecid-resistant viral mutants. In the last sentence (line 393-396), the authors speculate that some NA mutations may also exhibit Probenecid-resistance, but it is not clear why this important issue, which could further compare Probenecid with Oseltamivir, was not pursued further.

 Minor: The references are probably not MDPI-formatted. Line 408 needs to corrected.

  • We appreciate the reviewer’s thoughts and observations. Probenecid-resistant viruses are unlikely as the drug does not target the virus but reduces OAT3 and MAP kinase phosphorylation in virally infected cells. In this study, we noted that probenecid or PTGE compounds were less effective against A/Mississippi/3/2001 H275Y, suggesting that NA mutations may affect probenecid sensitivity, but the mechanism for this observation is currently unknown.
  • Thank you for catching the incomplete reference. Reference 7 has been corrected.

Reviewer 3 Report

Comments and Suggestions for Authors

The study lacks significant data (in vitro and in vivo toxicity of probenecid, in vivo efficacy, in vitro selectivity, etc.) and associated methods. Therefore, it is not possible to evaluate this study.

Author Response

We include responses to Reviewers 2 and 3. Reviewer 2 responses have not been changed, but Reviewer 3 responses are now included.

We have attached a revised manuscript that addresses the reviewer's comments.  Yellow highlights were the original responses, blue highlights were the second response, and gray represent the most recent. We tried to address Reviewer 3’s comments that,  "The study lacks significant data (in vitro and in vivo toxicity of probenecid, in vivo efficacy, in vitro selectivity, etc.) and associated methods. The responses are below:

Response to Reviewer 2–

We have highlighted additional edits in light blue.

  1. Authors' affiliation. None of the authors indicate affiliation. They
    provided only emails and one phone number. I would like to clarify the
    affiliations to the place the work was done (University? Private? Government?)
  • The affiliations of all authors have been added, and the place of work is noted as the University of Georgia by light blue highlights.

  1. The text needs to be written in a standard research language. "uM" is
    understandable but not correct to indicate micromolar concentrations. Kindly
    use the standard letter mu (μM).
  • We have made the modification as requested.

  1. What is this supposed to mean (line 260): 0.010.6 uM/625 uM?
  • The IC50/IC90 for prophylactic treatment of A549 cells with oseltamivir was 0.010.6 mM /625 m We hope this clarifies your question.

  1. Day 5 pi? What is it supposed to mean? It is said several times "day 5
    pi", kindly clarify in the text.
  • It is not clear what the question is. ‘Day 5 pi’ describes the time of treatment (lines 162, 312, 326), when the lungs were harvested post-infection (lines 166, 320), or the last point in body weight determination (line 342).

  1. Materials and Methods, indicate catalog numbers of reagents used. Make
    sure that the methods can be reproduced based on the author's description.
  • We have followed the journal preference which is to indicate the manufacturer, city, and state as requested.

  1. Overall, kindly proofread the manuscript. The final proofreading will help
    but might not catch all the issues.
  • Done

Response to Reviewer 3-

Reviewer 3 stated, “The study lacks significant data (in vitro and in vivo toxicity of probenecid, in vivo efficacy, in vitro selectivity, etc.) and associated methods. Therefore, it is not possible to evaluate this study”.

  • We disagree. In the introduction and throughout the revised manuscript, published results showing in vitro and in vivo probenecid efficacy are shown. In our manuscript, it is noted that probenecid is an FDA-approved drug with no in vitro or in vivo toxicity. Probenecid has been shown to have in vitro and in vivo antiviral activity for all influenza viruses, SARS-CoV-2 viruses, and respiratory syncytial virus (RSV) strains and variants tested, and safety data from a Phase II human COVID-19 dose-ranging study has shown a lack of clinical toxicity. Please see the related references below:
  1. PMID: 37515194, Oral Probenecid for Nonhospitalized Adults with Symptomatic Mild-to-Moderate COVID-19. Martin DE, Pandey N, Chavda P, Singh G, Sutariya R, Sancilio F, Tripp RA. Viruses. 2023 Jul 6;15(7):1508. doi: 10.3390/v15071508.
  2. PMID: 35632652, Probenecid Inhibits Respiratory Syncytial Virus (RSV) Replication. Murray J, Bergeron HC, Jones LP, Reener ZB, Martin DE, Sancilio FD, Tripp RA. Viruses. 2022 Apr 27;14(5):912. doi: 10.3390/v14050912.
  3. PMID: 35337018, Repurposing Probenecid to Inhibit SARS-CoV-2, Influenza Virus, and Respiratory Syncytial Virus (RSV) Replication. Tripp RA, Martin DE. Viruses. 2022 Mar 15;14(3):612. doi: 10.3390/v14030612.
  4. PMID: 34508172, Probenecid inhibits SARS-CoV-2 replication in vivo and in vitro. Murray J, Hogan RJ, Martin DE, Blahunka K, Sancilio FD, Balyan R, Lovern M, Still R,
  5. PMID: 23129053, Targeting organic anion transporter 3 with probenecid as a novel anti-influenza a virus strategy. Perwitasari O, Yan X, Johnson S, White C, Brooks P, Tompkins SM, Tripp RA. Antimicrob Agents Chemother. 2013 Jan;57(1):475-83. doi: 10.1128/AAC.01532-12. Epub 2012 Nov 5.

Round 2

Reviewer 2 Report

Comments and Suggestions for Authors

Some of my comments were responded to.

Author Response

Response to Reviewer –

We have highlighted additional edits in light blue.

  1. Authors' affiliation. None of the authors indicate affiliation. They
    provided only emails and one phone number. I would like to clarify the
    affiliations to the place the work was done (University? Private? Government?)
  • The affiliations of all authors have been added, and the place of work is noted as the University of Georgia by light blue highlights.
  1. The text needs to be written in a standard research language. "uM" is
    understandable but not correct to indicate micromolar concentrations. Kindly
    use the standard letter mu (μM).
  • We have made the modification as requested.
  1. What is this supposed to mean (line 260): 0.010.6 uM/625 uM?
  • The IC50/IC90 for prophylactic treatment of A549 cells with oseltamivir was 0.010.6 mM /625 m We hope this clarifies your question.

  1. Day 5 pi? What is it supposed to mean? It is said several times "day 5
    pi", kindly clarify in the text.
  • It is not clear what the question is. ‘Day 5 pi’ describes the time of treatment (lines 162, 312, 326), when the lungs were harvested post-infection (lines 166, 320), or the last point in body weight determination (line 342).
  1. Materials and Methods, indicate catalog numbers of reagents used. Make
    sure that the methods can be reproduced based on the author's description.
  • We have followed the journal preference which is to indicate the manufacturer, city, and state as requested.
  1. Overall, kindly proofread the manuscript. The final proofreading will help
    but might not catch all the issues.
  • done